# Reproducibility Report: Your Classifier is Secretly an Energy Based Model and You Should Treat it Like One

## Reproducibility Summary

**Scope of Reproducibility**

We validated the Joint Energy-Based model (JEM) training technique, recently developed by Grathwohl et al. [1]. Specifically, we checked performance on image classification, generation, and uncertainty calibration.

**Methodology**

We re-implemented the paper's pipeline from scratch, based on the algorithm described in the paper. We only referred to the authors' code for subtleties such as data normalization which were not explicitly mentioned in the paper. Training JEM took about 12 hours using a Wide ResNet 28-2 architecture [2].

**Results**

We verified that JEM performed similarly to how it was presented in [1]. We could not reproduce the exact numerical results of the paper due to the long training time of the algorithm.

**What was easy**

The paper is well-written, with the algorithm and motivation clearly explained.

**What was difficult**

We were not able to reproduce the exact results of the paper since constraints on computation time forced us to use a smaller network than the authors. Running the authors' method with the model and hyperparameters they described in their original paper would have required about 80 hours to train on our hardware (which we believe is comparable to the authors' hardware), rather than the 36 hours the authors reported in the paper.

While the training method produces a well-calibrated hybrid model, training itself is unstable. We needed to restart training a few times due to the loss diverging.

**Communication with original authors**

We spoke to the authors who corroborated the second of the above difficulties, however the origin of the lengthy training time is less clear.

## 1 Introduction

Research in generative models has historically been motivated by the potential benefit to tasks such as semisupervised learning, imputing missing data and/or data augmentation for discriminitive tasks, and calibrating uncertainty in predictive modelling. However, much of the recent literature in the subject is heavily motivated by generated sample quality. At present, there is a large gap in performance between generative modelling and downstream discriminative tasks. This is motivated, at least in part, by stark contrast between generative and discriminative deep neural network architectures.

The paper we reproduce in this report [1] bridges the gap between discrimination and generation by using a novel energy-based approach. Typically discriminative classifiers define a conditional probability distribution $p(y|\mathbf{x})$ over the labels $y \in \mathbb{R}^K$ (for $K$ classes), where (e.g. for images) $\mathbf{x} \in \mathbb{R}^{c \times h \times w}$ (channels, height, width). However, by reinterpreting of the pre-softmax network outputs of a neural network, the authors of [1] define a method to model the *data* distribution $p_\theta(\mathbf{x})$ as

$$p_\theta(\mathbf{x}) = \frac{\exp(-E_\theta(\mathbf{x}))}{Z(\theta)} \tag{1}$$

where $E_\theta$, known as the *energy function*, depends on the pre-softmax network outputs and continuously maps each image to a scalar. The normalization $Z(\theta) \equiv \int_{\mathbf{x}} \exp(-E_\theta(\mathbf{x}))$ is known as the *partition function*. Performing maximum likelihood estimation by differentiating (1) with respect to $\theta$ yields

$$\frac{\partial \log p_\theta(\mathbf{x})}{\partial \theta} = \mathbb{E}_{p_\theta(\mathbf{x}')}\left[\frac{\partial E_\theta(\mathbf{x}')}{\partial \theta}\right] - \frac{\partial E_\theta(\mathbf{x})}{\partial \theta}. \tag{2}$$

The authors approximate the intractable expectation in 2 by *stochastic gradient langevin descent* (SGLD) [3], a type of Markov chain Monte Carlo (MCMC) algorithm. We refer the reader to the references (particularly [1], [3]) for details on how this is implemented in practice. The full likelihood function that we must optimize with respect to the network parameters $\theta$ is then given summing the generative and discriminative contributions:

$$\mathcal{L} \equiv \log p_\theta(\mathbf{x}) + \log(p_\theta(y|\mathbf{x}). \tag{3}$$

For the purpose of this reproducibility study, we chose to replicate Algorithm 1 of [1] described in Appendix A of the paper. This is the primary novel algorithm proposed, known as *joint energy-based* model (JEM) training for jointly training a discriminative/generative (*hybrid*) model. This training scheme is a subset of the class *hybrid* model training methods (hybrid due to the joint generative/discriminative nature). The paper shows that JEM outperforms state-of-the-art hybrid models at both generative as well as discriminative modeling.

*NB: If we mention "the paper" without citation, we are referring to [1].*

## 2 Scope of reproducibility

We verify the following key claims made in the paper:

1. We verify the validity of the JEM-based training approach by implementing the model and training pipeline from scratch. Due to issues with training time, we could not verify the results for the exact ResNet architecture described in the paper (see Section 5 section for details). However, we trained a Wide ResNet (WRN) 28-2 (scaled down from the WRN 28-10 trained in the paper) and produced comparable generative and discriminative performance.

2. We verify that training with JEM improves uncertainty calibration compared to standard cross entropy minimization.

## 3 Methodology

We made every effort to perform an independent reproduction relying on the author's paper, however we occasionally had to refer to the author's code to obtain information not presented in the paper, particularly image preprocessing hyperparameters and the train/test split.

### 3.1 Model descriptions

While the paper used a Wide Resnet (WRN) 28-10, we scaled down to a WRN 28-2 for reasons of training time. (See Sec 5 for more details). We did not implement the WRN architectures ourselves, but found an existing implementation (based on this code) which we used.

### 3.2 Datasets

We used CIFAR10 [4] for all experiments, as in the original paper.

### 3.3 Hyperparameters

To train the WRN 28-2 we used the same hyperparameters that the paper used to train the WRN 28-10. While we did not have time to optimize hyperparameters for our scaled down network, we found that we were able to produce comparable results to what was found in the paper. Full hyperparameters are given in Table 1.

| hyperparameter | value |
|---|---|
| WRN width | 28 |
| WRN depth | 2 |
| initial LR | 0.0001 |
| train epochs | 150 |
| LR decay | 0.3 |
| LR decay epochs | 50, 100 |
| SGLD steps | 20 |
| buffer size | 10000 |
| buffer reinit freq | 0.05 |
| SGLD step size | 1 |
| SGLD noise | 0.01 |

Table 1: Hyperparameters used for training JEM reproduction.

### 3.4 Experimental setup and computational requirements

Our code is available here.

All experiments were run on an AWS p3.2xlarge instance with 8 CPUs and 1 NVIDIA V100 GPU. Training took about 12 hours for JEM hybrid training and about 2 hours to train an ordinary classifier using the hyperparameters given in Table 1. Before we scaled down our model architecture, we tried training a WRN 28-10 on an AWS p2.xlarge instance with 4 CPUs and an NVIDIA K80 GPU; this took about 40 hours.

As a time test, we ran the authors' code on a AWS p3.2xlarge instance. However we projected a training time of about 80 hours (as opposed to 36 as they described in their paper) based on time per iteration. This is why we decided to scale down our architecture.

## 4 Results

We trained a WRN 28-2 architecture using the hyperparameters described in section 3.3. In particular, we trained both a pure classifier as well as a hybrid model using the same architecture. In analogy with lines 5 and 6 of Table 1 of the paper, we report both the accuracy of the pure discriminative performance as well as the accuracy and inception score (IS) [5] of the hybrid model. We also went beyond the paper by generating samples from uniform noise in 4.3, and computed their inception scores as well. To compute the inception score, we adapt the code from here (code for ref [6]).

### 4.1 Result 1: Accuracy and Generative Samples

Table 2 compares the accuracy of the two architectures, where the 28-2 is our result and the 28-10 is taken from Table 1 of the paper. Table 3 compares both the generative and discriminative performances of the two architectures trained

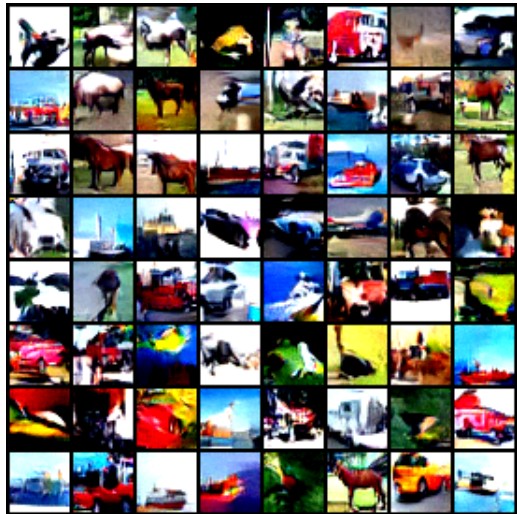

Figure 1: Samples from last checkpointed training buffer, top 64 samples with most confident predictions.

using JEM. The inception scores are computed using the top 10% of most confident inception-network-classified samples from the training buffer.

In our training pipeline, we saved checkpoints every 5 epochs. We also did a further study of the inception scores computed from (a) the final checkpointed training buffer (epoch 145), and (b) the ensembled last 5 training buffers between epochs 125 and 145. For both of these, we computed the inception score over the top $k$ most confident samples (according to the inception network) for $k \in \{100, 500, 1000, 5000, 10000\}$. The results for these are shown in Tables 4 and 5 respectively. Figure 1 shows the 64 most confident (according to the inception network) samples taken from the last saved checkpointed buffer.

| Architecture | Accuracy |
|---|---|
| WRN 28-2 (ours) | 89.59% |
| WRN 28-10 (paper) | 95.80% |

Table 2: Accuracy of the Wide resnet architecture trained purely as a classifier, comparison between 28-2 (ours) and 28-10 (paper).

| Architecture | Accuracy | Inception Score |
|---|---|---|
| WRN 28-2 (ours) | 83.72% | 19.39 |
| WRN 28-10 (paper) | 92.90% | 7.82 |

Table 3: Accuracy and inception scores of the Wide resnet architecture trained trained as a hybrid model using JEM training. For the inception score, we use the top 10% most confident predictions according to the inception network from the final checkpointed training buffer.

## 4.2 Result 2: Calibration

The *expected calibration error* (ECE) is a method to measure how well a model is calibrated – the extent to which a model's confidence predicts the true likelihood. Given training samples $\mathbf{x}_i$, a classifier's *confidence* on an example is defined as $\max_y p(y|\mathbf{x}_i)$. Binning these confidences into equally spaced buckets $\{B_m\}_{m=1}^{M}$, we can compute the *accuracy* as well as *confidence* of the model evaluated on all examples in $B_m$ for each $m$. These are given by

| $k$ | 100 | 500 | 1000 | 5000 | 10000 (Full) |
|---|---|---|---|---|---|
| **Inception Score** | 14.24 | 22.00 | 19.39 | 8.47 | 5.16 |

Table 4: *Inception scores of the last training checkpoint.* Inception scores of the top $k$ most confident predictions in the last checkpointed training buffer, for $k \in \{100, 500, 1000, 5000, 10000\}$.

| $k$ | 100 | 500 | 1000 | 5000 | 10000 (Full) |
|---|---|---|---|---|---|
| **Inception Score** | 12.00 | 13.45 | 10.67 | 4.37 | 2.88 |

Table 5: *Inception scores of computed over the ensembled last 5 checkpoints.* Inception scores of the top $k$ most confident predictions in the averaged last 5 checkpointed training buffers, for $k \in \{100, 500, 1000, 5000, 10000\}$.

$$\text{acc}(B_m) = \frac{1}{|B_m|} \sum_{i \in B_m} 1(\hat{y}_i = y_i) \tag{4}$$

$$\text{conf}(B_m) = \frac{1}{|B_m|} \sum_{i \in B_m} \hat{p}_i \tag{5}$$

where $\hat{y}_i$ and $\hat{p}_i$ are respectively the prediction and confidence of each example. The ECE is then given by

$$\text{ECE} = \sum_{m=1}^{m} \frac{|B_m|}{n} |\text{acc}(B_m) - \text{conf}(B_m)|. \tag{6}$$

An ideally calibrated model has an ECE of zero. We provide calibration plots for both ordinary supervised training as well as JEM for the WRN 28-2 architecture in Figure 2.

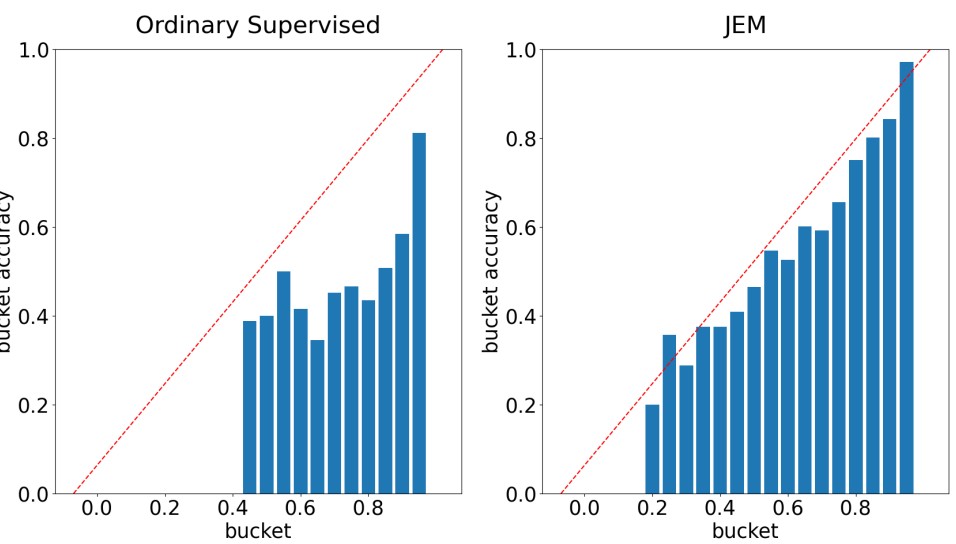

Figure 2: Calibration plots for ordinary supervised training (left) and JEM training (right). ECEs are 7.78% (supervised) and 4.31% (JEM). Dashed red line corresponds to perfect calibration.

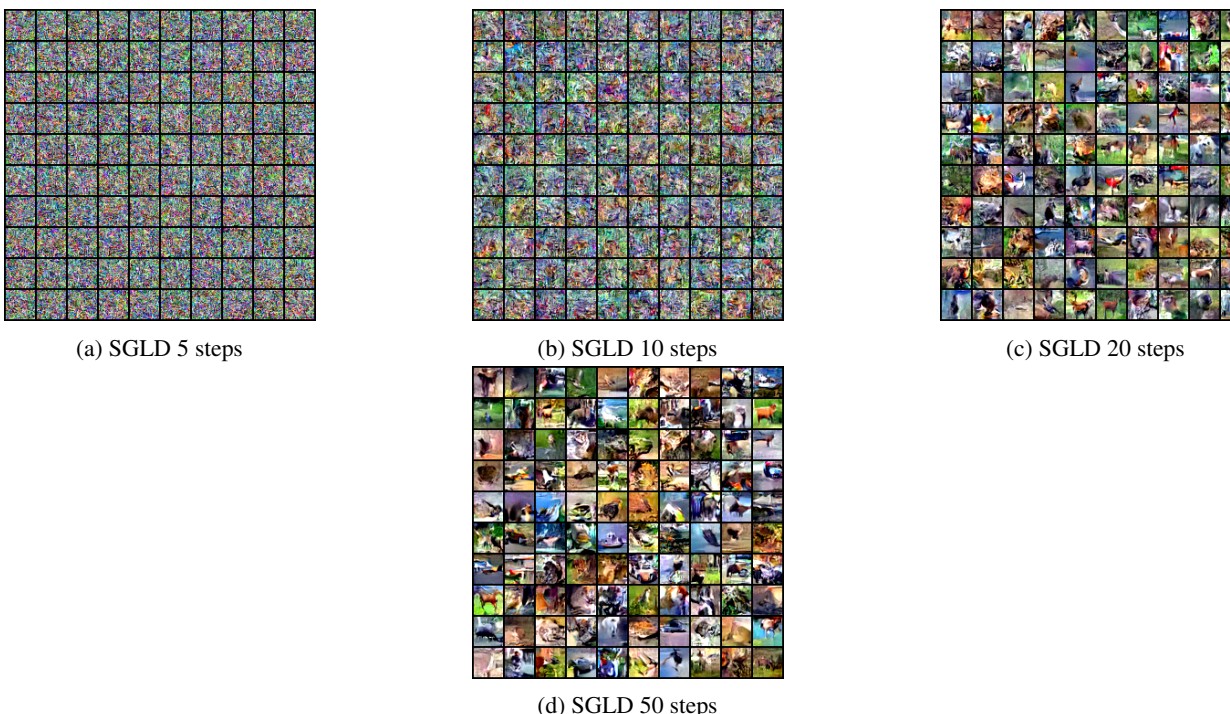

(a) SGLD 5 steps     (b) SGLD 10 steps     (c) SGLD 20 steps

(d) SGLD 50 steps

Figure 3: Fresh generated SGLD samples from uniformly random initialized noise.

### 4.3  Additional results not present in the original paper

We note that that in the original paper, "generated" samples were drawn from the training buffer of the last saved checkpoint, rather than having images generated from scratch. To emulate a true generative model, we created a "buffer" of size 100, initializing it with random uniform noise over $[-1, 1]$. We then evolved it for 5, 10, 20, 50, and 100 SGLD steps to see how well the model could generate images from scratch. See Figure 3 for generated samples at these evolution steps.

For reference, we also compute the inception scores of each of SGLD-evolved generated samples, which are given in Table 6.

| $k$ | 5 | 10 | 20 | 50 | 100 |
|---|---|---|---|---|---|
| **Inception Score** | 1.19 | 1.31 | 3.41 | 3.99 | 4.21 |

Table 6: Inception scores corresponding to the generated samples in Figure 3. Scores are computed using the 100 generated images at each (bold) step shown above.

## 5  Discussion

Our results generally tend to corroborate the validity of the JEM method to train hybrid generative/discriminative models. In particular, we were able to achieve comparable discriminative accuracy according to Table 3, and the samples we generated both from the final training buffer (Figure 1) as well as from uniform random noise (Figure 3) produced slightly unusual but decent-quality images.

We, however, note that the inception scores we obtained for the generated images were puzzling, as we would expect them to be lower than what was obtained in the paper (appendix B, table 5 (unconditional) – 7.82 for the last checkpoint and 7.79 for the ensemble). However we obtain larger scores; it seems odd that a less complex model would produce seemingly higher quality images. One possible source of this discrepancy is the way in we calculated the scores. We picked the images with the top $k$ most confident predictions according to the *inception* network, rather than with respect

to the WRN architecture we trained. Looking above Table 5 in Appendix B of the paper, we notice that they say that they compute the inception score by "keeping the top 10 percentile samples with highest $p(y|\mathbf{x})$ values". We assume that $p(y|\mathbf{x})$ refers to the network they trained, and not inception.

What's further interesting is that the images generated from uniform random noise have significantly lower inception scores than those sampled from the training buffer of the last checkpoint. We hypothesize that this is because the training buffer images evolve many more SGLD steps over the course of training. While the generated images qualitatively look good, we plan to run more experiments with different architectures in the next iteration to understand the distribution of the inception score better.

## 5.1 What was easy

This paper was well-written, and the training algorithm (Algorithm 1 of Appendix A of [1]) was clearly described so we were able to implement it almost directly from the paper.

## 5.2 What was difficult

Due to runtime, we found that we were not able to exactly replicate the results of the paper – we found that running the authors' original code on the p3 instance would have taken about 80 hours to train a WRN-28-10 model, rather than 36 as they'd said in their paper. However, even though we used this training method on a scaled-down model (WRN 28-2 rather than WRN 28-10 like done in the paper), we feel that the method does generalize to other architectures.

In the paper [1], the authors claimed to be able to train a WRN-28-10 model using the JEM training method in about 36 hours on a single GPU. However, even when we ran their code on our hardware (mentioned in Sec 3.4), we extrapolated that it would take at least 80 hours to train a full-scale 28-10 model. Hence, for this initial reproducibility, we decided to scale down the model architecture while trying to test the efficacy of the method.

One other subtle difficulty which we encountered was that while the result of training neural networks with the JEM method produces a well-performing hybrid model, the training itself is highly unstable. The JEM loss function (eqs. (2), (8) of [1]) is prone to diverging, hence requiring restarting training from the most recent saved checkpoint. While this was mentioned in Appendix H3 of [1], we felt that this could have been stated and explored more clearly.

## 5.3 Communication with original authors

We did communicate with the original authors, who corroborated the second of the above difficulties described. The origin of the lengthy training time, however, is less clear.

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
