# OpenReview forum: "Reproducibility Report: Your Classifier is Secretly an Energy Based Model and You Should Treat it Like One"
_ML_Reproducibility_Challenge/2020 — Reject_

### Official Review · AnonReviewer2 · 2021-02-27
**Review. Interesting Replication**

**Rating:** 7
**Confidence:** 5

**Review:**

Summary: This is a replication of the ICLR paper "Your Classifier is Secretly an Energy Based Model and You Should Treat it Like On" by Grathwohl  et al. (2020). This replicability-report focuses on studying the results both on accuracy and model calibration of the original paper. The report seems to corroborate the results of the Joint Energy-Based Model (JEM) and provides interesting results on the inception scores obtained.

With respect to the pros, this replicability study is clear, well organized, easy to read, and seem to study the paper in great detail considering the limited time. The report also contains all the main expected elements for the Replicability Challenge, including the type of contact the author of the report had with the authors of the original paper, scope, code, computational platform, difference of the hyperparameter in the original paper vs. the replicability study, the implication (discussion) of the results, limitations, and additional insights. This report has a clearly delineated summary.

On the other hand, there are a few elements of the report that could have been clarified a little more. For instance, the authors claimed they performed additional experiments with uniform noise and computed its scores - in the words of the authors this evaluation went beyond what was reported in the original ICLR paper. However, the original ICLR paper did consider uniform noise -- see section F.2. Clarification on this can improve the report's clarity.

**Familiar With The Original Paper:**

I have read the original paper

**Reproducibility Summary:**

Report has summary

---

### Official Review · AnonReviewer1 · 2021-03-01
**Questionable experiments, incorrect results**

**Rating:** 3
**Confidence:** 4

**Review:**

I am puzzled by the authors' choice to reproduce the results of the paper without looking at their code while only looking at their code when needed. Why not just use the provided code and build upon it?

The authors state that the limited computational resources were a bottleneck in their reproduction, which is fair. However, given such a limitation one would expect experiments to be chosen more carefully with a directed purpose, but this does not seem to be the case. For example, Table 2 seems pointless to me. The focus of the paper is the joint energy-based models (JEM) and the 95.8% purely discriminator accuracy was presented only to provide a point of reference. Why spend time on this?

Also, Table 3 is puzzling. The authors achieve a better inception score, far better than everything else (including purely generative models) listed in Table 1 of the original paper. This would be a surprising result that requires an explanation or a discussion, but the authors do not acknowledge it at all. Well, the inception score with CIFAR10 (10 classes) is by definition <=10, so the reported 19.39 must be an incorrect number. In fact Tables 4 and 5 have the same issue.

I also do not find the other experiments to provide much new information. For these reasons, I recommend the reproducibility report be rejected.

Errata:
1. line 52, Due 'to' issues with training time
2. The reference [3] https://www.cs.toronto.edu/ kriz/cifar.html. is not accessible.
3. line 101, the right parenthesis of max_y p(y|x_i)
4. The reference for wide resnet should be included.

**Familiar With The Original Paper:**

I have read the original paper

**Reproducibility Summary:**

Report has summary

---

### Official Review · AnonReviewer3 · 2021-03-02
**Fair reproduction of the results**

**Rating:** 7
**Confidence:** 3

**Review:**

1. The authors clearly state the scope and limitation of the reproduction experiments.
2. The results are “reproduced" using a scaled down architecture (Wide ResNet 28-2 instead of Wide ResNet 28-10). The authors claim that the disparity in reported training time in the original paper and the authors’ own estimate is the reason for scaling down the architecture.
3. Only one aspect of the results presented in the original paper is reproduced. However, the authors present fresh generative samples from noise and show how the generative samples evolve over SGLD steps.
4. The authors have used exactly the same hyperparameters used in the original paper. The authors mention that the hyperparameters may not have been optimal for the downgraded architecture used for reproduction of results.
5. The authors have written the code from scratch (repository made public), but refer to the original code only for a few pre-processing steps that are not clear in the original paper.
6. The authors have communicated with the original authors for clarifications on training instabilities and training time.


7. Typos:
	* Line 101: Missing closing parentheses
	* Line 108: a buffer, instead of an buffer




**Familiar With The Original Paper:**

I have read the original paper

**Reproducibility Summary:**

Report has summary

---

### Decision · Program_Chairs · 2021-03-31

**Decision:**

Reject

**Comment:**

While starting the code base from scratch is an interesting contribution, the reproducibility of the original paper is not carried out extensively enough.